# Some Preliminary Results to Eradicate Leukemic Cells in Extracorporeal Circulation by Actuating Doxorubicin-Loaded Nanochains of Fe_3_O_4_ Nanoparticles

**DOI:** 10.3390/cells11132007

**Published:** 2022-06-23

**Authors:** Xiawen Zheng, Xiaoli Mai, Siyuan Bao, Peng Wang, Yu Hong, Yuexia Han, Jianfei Sun, Fei Xiong

**Affiliations:** 1The State Key Laboratory of Bioelectronics, Jiangsu Key Laboratory of Biomaterials and Devices, School of Biological Sciences and Medical Engineering, Southeast University, Dingjiaqiao 87, Nanjing 210009, China; wuzhibluelove@126.com (X.Z.); bsy980926@163.com (S.B.); 2Drum Tower Hospital Affiliated to Medical School of Nanjing University, Zhongshan Road 321, Nanjing 210008, China; maixl@njglyy.com (X.M.); peng_wang0324@163.com (P.W.); 3The State Key Laboratory of Natural Medicine, Department of Traditional Chinese Medicine, College of Chinese Materia Medica, China Pharmaceutical University, Nanjing 210009, China; yhongwork@163.com; 4Technology and Engineering Center for Space Utilization, Chinese Academy of Sciences, Beijing 100094, China

**Keywords:** nanochains, leukemia, drug carrier, rotational magnetic field, extracorporeal circulation

## Abstract

Leukemia is a non-solid cancer which features the malignant proliferation of leukocytes. Excessive leukocytes of lesions in peripheral blood will infiltrate organs, resulting in intumescence and weakening treatment efficiency. In this study, we proposed a novel approach for targeted clearance of the leukocytes in the peripheral blood ex vivo, which employed magnetic nanochains to selectively destroy the leukocytes of the lesions. The nanochains were doxorubicin-loaded nanochains of Fe_3_O_4_ nanoparticles which were fabricated by the solvent exchange method combined with magnetic field-directed self-assembly. Firstly, the nanochains were added into the peripheral blood during extracorporeal circulation and subjected to a rotational magnetic field for actuation. The leukocytes of the lesion were then conjugated by the nanochains via folic acid (FA) targeting. Finally, the rotational magnetic field actuated the nanochains to release the drugs and effectively damage the cytomembrane of the leukocytes. This strategy was conceptually shown in vitro (K562 cell line) and the method’s safety was evaluated in a rat model. The preliminary results demonstrate that the nanochains are biocompatible and suitable as drug carriers, showing direct lethal action to the leukemic cells combined with a rotational magnetic field. More importantly to note is that the nanochains can be effectively kept from entry into the body. We believe this extracorporeal circulation-based strategy by activating nanochains magnetically could serve as a potential method for leukemia treatment in the future.

## 1. Introduction

Leukemia is a common non-solid cancer which features the incomplete differentiation of naive leukocytes in bone marrow and the malignant proliferation of leukocytes [1,2,3]. This disease currently poses a major threat to human health and the mortality rate remains high. Although pathogenesis of the leukemia lies in the bone marrow, the excessive leukocytes of lesions in peripheral blood can cause inhibition of hematopoietic function, infiltration into other organs such as the liver, spleen and the lymph nodes, and organomegaly [4,5]. This will worsen the body function of patients and greatly weaken treatment efficiency [6]. Thus, clearance of the cancerous leukocytes in the peripheral blood is pivotal during the treatment of leukemia. Given the side effects of chemotherapeutic drugs, it would be desirable to precisely act upon the cancerous leukocytes rather than on the circulation of the body. For this purpose, diverse targeted drug delivery systems have been developed relying on passive or active targeting [7,8]. To date, there is increasing interest in controllable drug release systems [9]. The chemical, physical and biological cues have been employed to control the drug release in different types of carriers [10,11,12]. Some advantages, such as the maximizing drug efficacy, the overcoming of drug resistance and the associated side effects, were shown [13,14,15]. However, the control of drug release with two or more means has seldom been reported.

However, there remain some aspects awaiting improvement, such as low targeting efficiency [16], short circulation time [17,18] and especially the questionable safety of nanomaterials in vivo [19]. Therefore, it is urgent and imperative to update the current targeting strategy for leukemia.

Magnetic nanomaterials have played an important role in this area due to the unique advantage of remote controllability by magnetic field, which combined the passive target and the active target [20,21,22]. The swimming behaviors of magnetic micro/nano assemblies actuated by the magnetic field had been summarized, which indicated that even with a permanent magnet in the laboratory, the magnetic assemblies above 100 nm can be controlled to move or rotate in viscous fluids [23]. In particular, the chain-like structure was thought to be advantageous because of good actuation, ease of fabrication and fluidic suitability [24]. Moreover, the magnetic nanomaterials have been found to be able to disrupt the cellular skeleton or membrane via magneto-mechanical force, which facilitated the release of loaded drugs and played a synergistic role with chemotherapy [25,26,27]. Here, even the nanochains with a size about 60 nm in lysosomes was found to show a significant disruptive effect by actuation with a rotating magnetic field of low frequency so that the programmed cellular death was induced [28,29]. However, as far as the leukemia is concerned, the common targeting strategies become unsuitable because the cancerous cells distribute throughout the body. Thus, the targeted clearance of leukemia cells in the peripheral blood remains a challenging task.

Here, we proposed and conceptually proved a novel strategy for this issue. The idea was initially derived from hemodialysis that is a common technique in clinic for uremia patients. In this strategy, the blood is diverted from the body and has an extracorporeal circulation. The DOX-loaded magnetic nanochains come into the blood in vitro and bond to the cancerous leukocytes via folic acid (FA) targeting that is overexpressed on cancerous cells [30,31,32,33]. During the circulation in vitro, the blood is subjected to a rotational magnetic field. Based on the above-mentioned cases, the rotational magnetic field will actuate the magnetic nanochains to rupture the organelles including the lysosomes and the cytoskeletons so that the cellular environment turns acidic, facilitating the release of DOX and the apoptosis of cancerous cells [34,35,36,37]. Moreover, another advantage of this strategy is that the magnetic assembly is facile to get prompt separation. In our previous study, the efficiency of magnetic separation can reach 90% in 5 min [38,39]. After killing the cancerous leukocytes in vitro, the magnetic nanochains could be effectively removed from entering body and collected for recycling, which greatly reduces the dosage and enhances the safety of magnetic nanomaterials in practical applications.

Given that the normal cells expressed little FA receptor, the majority of magnetic nanochains cannot bind to the normal cells. There may only be a small quantity of nanochains that adhere to surface of the normal cells. This adherence was very weak, which will yield insignificant force on the normal cells in the presence of the rotational magnetic field.

It was known that DOX is an effective anti-tumor drug whose mechanism is clear, and the loading of drugs on magnetic nanomaterials is well developed. The novelty lies in the extracorporeal circulation that is potentially innovative in its application. We believe this strategy could be safe and promising for the clinical translation of leukemia treatment.

## 2. Materials and Methods

### 2.1. Synthesis of Polyethylene Glycol-Polylactic Acid-P-Nitrobenzoate (PEG-PLA-NPC)

1.6 g of PEG-PLA was added to 20 mL of dichloromethane to dissolve by shaking and ultrasonic vibration. It was then put into a two-neck flask. 80 μL of triethylamine was added after filling with nitrogen and ice bath. 112 mg of p-nitrophenyl chloroformate (P-NPC) was dissolved in 5 mL of dichloromethane and slowly dripped from the side mouth of the two-necked flask with a syringe for 30 min. After reacting in an ice bath for 1 h and with stirring at room temperature for 24 h, it was stored in a 25 mL flask and concentrated at 45 °C to 1–2 mL by rotary evaporation. It was then transferred to a 50 mL beaker, with the frozen ether added slowly while stirring. After complete precipitation, the supernatant was filtered off and washed three times to remove excessive p-NPC. Vacuum drying was kept at 25 °C for 6 h to obtain PEG-PLA-NPC.

### 2.2. Synthesis of Polyethylene Glycol-Polylactic Acid-Hydrazine (PEG-PLA-Hyd)

800.39 mg of PEG-PLA-NPC was added to 10 mL of dichloromethane to dissolve by shaking and ultrasonic vibration. It was then transferred to a two-neck flask. 63 μL of hydrazine hydrate was added after filling with nitrogen and ice bath. The molar ratio of PEG-PLA-NPC to hydrazine hydrate was 1:10. After reacting in an ice bath for 1 h and stirring at room temperature for 24 h, it was stored in a 25 mL flask and concentrated at 45 °C to 1–2 mL by rotary evaporation. The next step involved placing it in a 50 mL beaker, with ice ether added slowly while stirring. After complete precipitation, the supernatant was filtered off and washed three times to remove excessive ether. It was quickly put in a vacuum drying oven and dried under vacuum at 25 °C for 6 h to obtain PEG-PLA-Hyd.

### 2.3. Synthesis of Polyethylene Glycol-Polylactic Acid-Hydrazone Bond-Doxorubicin (PEG-PLA-Dox)

355.26 mg of PEG-PLA-Hyd was added to 8 mL DMF to dissolve ultrasonically with shaking. 21.82 mg of doxorubicin hydrochloride was added to 4 mL DMF for ultrasonic dispersion. The two solutions were mixed and placed in a 50 mL two-neck flask. 15 μL triethylamine was added, and the reaction was stirred for 24 h under the conditions of nitrogen gas and 60 °C oil bath in the dark. PEG-PLA-Dox was obtained after freeze-drying at −80 °C.

### 2.4. Synthesis of Polylactic Acid-Polyethylene Glycol-Folic Acid (PLA-PEG-FA)

200.96 mg of PLA-PEG-COOH was added to 8 mL of DMSO and magnetically stirred for 4 h to dissolve. It was then transferred into a 50 mL Erlenmeyer flask. 24.23 mg of EDC was added to the Erlenmeyer flask, and after 5 min of dissolution, 8.35 mg of NHS was added. The molar ratio of PLA-PEG-COOH: EDC: NHS is 1:5:2.5. It was protected from light to activate for 5 h. Eight mL of DMSO was added to the Erlenmeyer flask, followed by the addition of 12.45 mg of folic acid to the DMSO while stirring in the dark until it dissolved. The activation solution was dropped into the folic acid solution, and the reaction was stirred for 24 h in the dark. After the reaction, it was transferred to a dialysis bag with a molecular weight cut-off of 2000 for dialysis for 72 h. PLA-PEG-FA was obtained after lyophilization at −80 °C.

### 2.5. Preparation of Targeted Dox-Loaded Magnetic Nanochains

2 mL of ethanol was added to the chloroform solution of 200 μL OA@Fe_3_O_4_, prepared by the Jiangsu Key Laboratory of Biomaterials and Devices [13]. The specific process is as follows. The OA-coated Fe_3_O_4_ nanoparticles were successfully synthesized via a thermal decomposition procedure. In a typical experiment, 2 mmol iron (III) acetylacetonate and 11 mmol oleic acid were added into a 100 mL three-necked flask with 20 mL benzyl ether as reaction solvent. The temperature of the reaction system was ramped to 220 °C at a heating rate of 3.3 °C/min. The reaction lasted for 1 h to form the nucleus of nanoparticles. Then the reaction system was heated at rate 3.3 °C/min to 290 °C, lasting for 30 min which was the growth stage of nanoparticles. Nitrogen gas was continuously introduced during the experiment to remove oxygen. The heat source was removed after the reaction was complete. The reaction system was then transferred to a beaker and washed with ethanol three times to clear off the residual oleic acid, benzyl ether, and unreacted precursors via magnetic separation. Finally, the magnetic OA@Fe_3_O_4_ nanoparticles were fixed in N-hexane and stored for use.

After magnetic adsorption for 2 min, the supernatant was discarded. 1 mL of tetrahydrofuran was added, and the mixture was ultrasonically dispersed in a water bath. Thirty mg of PLA-PEG, 10 mg PLA-PEG-FA, 10 mg PEG-PLA-Dox was added in a water bath until completely dissolved. Twenty mg of F68 was added to co-dissolve, and then a syringe was used to quickly inject the above solution into 9 mL of water. After that, ultrasonic vibration was employed for 5 min at a rated power of 750 W. It is also important to note that it vibrates every two seconds followed by a two-second pause. The above solution was transferred to a vial, which was heated in a 40 °C water bath under an 80 mT magnetic field to volatilize the organic solvent. It was then transferred to a dialysis bag with a molecular weight cut-off of 12,000–14,000, dialyzed with distilled water for 24 h, and filtered with a 0.22 μm microporous membrane. It was washed three times with an ultrafiltration tube and concentrated, and targeted Dox-loaded magnetic nanochains were obtained.

### 2.6. Characterization of OA@Fe_3_O_4_ Nanoparticles and Magnetic Nanochains

The morphology of OA@Fe_3_O_4_ nanoparticles and magnetic nanochains was observed by transmission electron microscopy (TEM). The particle size and Zeta potential of the magnetic nanochains were measured using a Malvern potential particle size analyzer at 25 °C. The hysteresis loop of the magnetic fluid sample was measured using a vibrating sample magnetometer. One milliliter of 1 mg/mL magnetic nanochains was added in water, pH 7.4 PBS and PBS with 10% fetal bovine serum at 37 °C, respectively. The hydrodynamic size was measured at 0 h, 8 h and 24 h postoperatively. Ten milliliters of 1 mg/mL Dox-loaded magnetic nanochains was placed in a dialysis bag and dialyzed against pH 5.0 and pH 7.0 PBS for 48 h at 37 °C, respectively. The drug release was measured by UV–vis spectroscopy at 0 h, 2 h, 4 h, 6 h, 12 h, 24 h and 48 h postoperatively.

### 2.7. Cell Culture

The culture of leukemia cell K562 (derived from the Chinese Academy of Sciences cell bank, Shanghai, China) was conducted in a 37 °C, 5% CO_2_ incubator using RPMI-1640 medium containing 10% FBS. The culture medium was changed every two days, and the cells could be passaged when the cells were about 90% confluent.

### 2.8. Cell Viability

A CCK-8 assay was used to measure cell viability. K562 cells in the logarithmic growth phase were seeded in a 96-well plate at 100 μL per well (about 4 × 10^5^ cells/mL), and 100 μL of MNCs and FA-MNCs of different concentrations diluted in the culture medium were added. The concentrations were 50, 100, 150, 200 μg/mL, and the edge holes were filled with sterile PBS and placed in a 37 °C, 5% CO_2_ cell incubator for 24 h. The 96-well plate was centrifuged at 1000 rpm for 10 min, and then the culture medium was aspirated. The CCK-8 solution was mixed in the whole medium at a ratio of 1:10, 100 μL of CCK-8 solution was added to each well, and the incubation was continued for 1 h. The shaker was shaken at high speed for 30 s, and the absorbance (OD) was measured at 450 nm on the microplate reader. In addition, it was applied for 1 h under a 30 Hz rotating magnetic field every day to investigate the damaging effect of the targeted magnetic nanochain on leukemia cells. It is worth mentioning that the equation of the cell survival rate is Cell Survival Rate = (As − Am − Ab)/(Ac − Ab) × 100%. Specifically, Ab is the OD value of the culture medium hole, Ac indicates the OD value of the control cell, As is the OD value of the magnetic nanocarrier cell, and Am refers to the OD value of the control magnetic nanocarrier.

In addition, the killing effect of targeted Dox-loaded magnetic nanochains on leukemia cells under a rotating magnetic field was investigated. The cell seeding was the same as above, which was divided into three groups: the targeted Dox-loaded magnetic nanochains with magnetic field action group (FA-MNCs-Dox (MF+)), the targeted Dox-loaded magnetic nanochains without magnetic field action group (FA-MNCs-Dox (MF-)), and the drug control group (Dox). The final concentration of iron was 50, 100, 200 μg/mL. The drug control group used a Dox solution (5 μg/mL) with the same release concentration of Dox in FA-MNCs-Dox. The remaining operations were described as above.

### 2.9. Calcein-AM/PI Double Staining

A Calcein-AM/PI Live/Dead Cell Dual Staining Kit (KeyGEN, Nanjing, China) was used to observe live and dead cells. In brief, K562 cells were seeded at 1 mL per well (approximately 5 × 10^5^ cells/mL). 1 mL of FA-MNCs, FA-MNCs-Dox of different concentrations diluted with the culture medium were added. The non-rotating magnetic field group was taken as the control group, the final concentration of iron was 200 μg/mL, and it was placed in a 37 °C, 5% CO_2_ cell incubator for 24 h. After applying it under a 30 Hz magnetic field for 1 h, the solution was aspirated and centrifuged at 2500 rpm for 5 min for collection. Cells were washed twice with a 1 × Assay Buffer. The cell suspension was prepared with 1×Assay Buffer to make the density of 1 × 10^5^–1 × 10^6^ cells/mL. 5 μL Calcein-AM solution and 15 μL PI solution were added to 5 mL 1×Assay Buffer. The staining working solution and the cell suspension were mixed at a ratio of 1:2 and placed under a fluorescence microscope for observation.

### 2.10. Simulation and Measurement of Rotating Magnetic Field

The multi-physics coupling in COMSOL Multiphysics was used to simulate the rotating magnetic field distribution. By setting constitutive parameters, magnet size and spacing, the influence of magnet parameters and spacing on the magnetic field distribution was investigated. Magnets with similar parameters were used to build a device with the same parameters as the simulated magnetic field. Teslameter was used to measure the true magnetic field strength and direction changes.

### 2.11. Blood Compatibility of Magnetic Nanochains—Hemolysis Test

In the experimental group, 2.5 mL of 2% red blood cell suspension which was the peripheral blood taken from the tail vein of the rats, 2.5 mL of normal saline and samples of different concentrations (50 μg/mL, 100 μg/mL, 200 μg/mL) were added to the test tube. The distilled water group was used as a positive control group, and the saline group was used as a negative control group. After mixing, they were immediately placed in a 37 °C ± 0.5 °C incubator. 100 μL of the supernatant was taken into a 96-well plate every 1 ho. Each well was filled with physiological saline to 200 μL. The absorbance was measured at 540 nm of the hemoglobin ultraviolet absorption peak with a microplate reader. The hemolysis rate Z = (A − A0)/(A1 − A0) × 100%, where the absorption value of the negative control group was A0, the absorption value of the positive control group was A1, and the sample group was A.

### 2.12. Coagulation Test

Samples of each group were dropped on the slide in advance. Among them, the diluted solution was divided into physiological saline and heparin sodium physiological saline, and the sample concentration was 50, 100, and 200 μg/mL. Blood was dropped onto the prepared slides under different test conditions. A clean needle was used to pick up the blood drop from the same direction every 10 s until the blood streak was picked up, then the blood began to coagulate. The clotting time of the three groups were recorded and the average value was taken.

### 2.13. Blood Routine Examination

All experimental animals were randomly divided into four groups, with each group having three animals. They were respectively treated by Extracorporeal Circulation (ECC group), Magnetic Nanoparticle Chain (MNC group), Statistic Magnetic Field (SMF group) and Rotating Magnetic Field (RMF group). After the surgery, 0.2 mL of arterial blood was collected before cardiopulmonary bypass and 1 h after cardiopulmonary bypass for blood routine examination.

### 2.14. Animal Experiments

All animal experiments were conducted following the guidelines of the Animal Research Ethics Board of Southeast University. Full details of the study approval can be found in the approval ID: 20080925. All of the animal experiments were performed in compliance with the Guidelines of the Animal Research Ethics Board of Southeast University. This ethics board also approved the animal studies.

## 3. Results

The concept of this strategy was schematically shown in Figure 1.

PLA-PEG coated Fe_3_O_4_ nanochains were fabricated by the magnetic field-directed assembly of oleic acid-capped Fe_3_O_4_ nanoparticles during solvent exchange. This process was schematically shown in Figure 2a. The PLA-PEG molecules, commercially available from Shandong Daigang Co., Ltd. (Shandong, China), can form a stable coating layer via hydrophobic interaction between the amphiphilic polymer and the oleic acid. The morphology and the hydrodynamic size of the original oleic acid@Fe_3_O_4_ nanoparticles are shown in Figure 2b and in the Appendix A, respectively, from which the nanoparticles can be seen to be highly uniform. The synthesized nanochains are shown in Figure 2c. The ξ potential and the hydrodynamic size of nanochains are shown in Appendix A, from which the nanochains can be seen as negatively charged (−16 mV) and relatively uniform. It was experimentally found that the polymer composition, the magnetic field intensity and the emulsification parameters had a great impact upon the fabrication of the nanochains. The addition of Tween 80 caused the nanochains to be more branched and longer, making the individual nanoparticle almost non-existent. However, the addition of F68 caused the nanochains to be shorter but uniform (Appendix A). Meanwhile, there was an optimized amount of polymer for the fabrication of nanochains (Appendix A). Too much polymer will self-assemble into micelles, hindering the formation of nanochains. Hence, the optimized ratio between PLA-PEG and OA@Fe_3_O_4_ nanoparticles was determined to be 2:50 (*w*/*w*). Given that the fabrication of nanochains was a self-assembly process during the magnetic field-directed emulsification, the magnetic field intensity and the emulsification time were also important. The experimental results are shown in Appendix A. In our experiments, 80 mT and 5 min were adopted to assemble the nanochains. The stability of the magnetic nanochains in biological buffers was also studied. The hydrodynamic size measurements were carried out in different media at different time points (0, 8 and 24 h). The results were shown in Figure 2c. It was found that there was a bit of agglomeration in the solutions of high ionic strength, but the hydrodynamic size remained below 220 nm, signifying that the leukocytes can also uptake the nanochains. In addition, the nanochains remained superparamagnetic (Appendix A). However, due to the assembly of individual units, the collective response for the external magnetic field was significantly augmented.

Doxorubicin was loaded into the polymer layer capping the nanochains resulting from the hydrophobic interaction. The loading concentration of drugs in nanochains was 1.2 wt.% (DOX/Fe). Quantification of DOX in nanochains was performed as follows. 0.5 mL DOX-loaded nanochains (Fe concentration: about 1 mg/mL) were sonicated in 1 mL methanol and centrifugated at 10,000 rpm for 10 min. The supernatant obtained was analyzed for drug loading in nanochains by using ultraviolet-visible spectrophotometry to test the DOX content.

The pH-dependent drug release was tested. The standard curve of DOX is shown in Appendix A, and the absorbance was linearly and positively correlated with DOX concentration. It was found that the drug release was significantly faster in the acidic condition than that in the neutral condition (Figure 2e). The total released amount of DOX in the neutral medium was 38.32 μg, while it was 72.06 μg in the acidic medium. Thus, it is important to pinpoint that this extracorporeal circulation-based approach of activating nanochains magnetically is useful in chemotherapy to eradicate cancerous cells. In addition, the biocompatibility of nanochains was also tested. A CCK-8 assay was used to examine the viability of K562 cells under the treatment of bare nanochains and FA-modified nanochains (Figure 2f). No matter whether the FA was modified, the cell viability was above 90% after 24 h of co-culture for both groups. Even for the Fe concentration of 200 μg/mL, there remained little cytotoxicity for both groups. Endocytosis of nanochains in both groups by the K562 cells was determined by Fe element quantitative measurement (Figure 2g). It can be seen that the nanochains largely entered the cells within 12 h, and the modification of FA significantly enhanced the endocytosis of the nanochains.

The rotational magnetic field was constructed by a pair of bar-shaped permanent magnets driven by a motor (Figure 3a). The packaged machine is shown in Appendix A. The field distribution of the pair of magnets were simulated and experimentally measured. A 3D model of the magnets is shown in Figure 3b. The stereogram of the simulation is shown in Appendix A. The simulated field distribution in cross-section is shown in Figure 3c. Seen from the simulation, the field intensity fleetly reduced from 146 mT on the surface of magnet into 70 mT in a position of 5 mm away from the surface. Thereafter, the field intensity was reduced from 70 mT into 30 mT, while the distance was increased from 5 mm to 20 mm, and the variation of field intensity tended to be constant. The field intensity in different positions were also experimentally measured, which can be seen in Figure 3d. In the subsequent experiments, the cells were put within the range of 5 mm to 15 mm for action by magnetic force.

We employed the co-incubation of K562 cells and the FA-nanochains to conceptually show feasibility of the nanochains-energized chemotherapy. The FA-nanochains were incubated with the K562 cells, which were subjected to the rotational magnetic field, for 1 h daily. After three days of treatment, the dosage dependent apoptosis of leukocytes was significantly promoted dependent (Figure 4a). This case can also be corroborated by the cellular morphology. With the increase of the amount of nanochains, the leukocytes were found to gradually break down to pieces, indicating the damaging action of the nanochains resulting from the rotational magnetic force (Figure 4b). The nanochains were found to significantly accumulate inside the cells and were wrapped by vesicles (Figure 4c). In the margin, the nanochains were sparse and showed slender morphology (Figure 4c Inset). In the absence of a magnetic force, the lysosomes were clear and regular, wrapping the magnetic nanochains inside (Figure 4d). After the action of the magnetic force, the edge of vesicles and lysosomes became blurred, leaving some black na-nomaterials outside. (Figure 4e). This case proved that the nanochains were actuated by the rotational magnetic field to destroy the organelles as well as to efficiently kill the cancerous cells.

Then, the half inhibitory concentration (IC50) value of DOX, FA-MNCs-Dox MF− and FA-MNCs-Dox MF+ for K562 cells were calculated based on the OD values of the CCK-8 double dilution method. The results were 2.72, 1.97 and 1.58 ug/mL, respectively, and they are the concentration of Dox and drug-loaded Dox. Meanwhile, we evaluated the effect of Dox-loaded nanochains on the cell viability of K562 upon magnetic response. As results, Dox-loaded FA-MNCs-Dox aggravated the cytotoxicity to K562 cells and expanded the decrease in cell viability, compared to that induced by Dox with the same dose as FA-MNCs-Dox. While MF did not appear to amplify this cytotoxic effect, we think this is because the loaded Dox is so toxic that it probably masks the effect of MF-induced on cell damage (Figure 4g).

Interestingly, the K562 cells were dual-stained by Calcein-AM and PI to show the cellular morphology more clearly. Seen from the microscopic fluorescent images (Figure 4f), the addition of FA-magnetic nanochains slightly augmented the red fluorescence, proving that the targeting by FA was feasible and effective. After the rotational magnetic field was imposed, the red fluorescent signals were significantly augmented, indicating that the permeability of the cytomembrane was enhanced by the mechanical action. This phenomenon proved that the FA-DOX-magnetic nanochains played a robot-like role in the increased apoptosis rate of the cancerous cells under actuation of the rotational magnetic field.

Considering the safety issue of drug nanocarriers in vivo, the magnetic nanochains had better circulation in vitro rather as opposed to entering into the human body. In our strategy, the blood will be collected from the body to form the extracorporeal circulation, and the na-nochains-based killing of cancerous cells took place in vitro. Thus, we preliminarily evaluated blood compatibility of the magnetic nanochains (Appendix A). The hemolysis experimental results showed that the hemolysis rate of each group was less than 5% within 3 h, indicating that the nanochains caused insignificant hemolysis for blood cells. Seen from the solidification time of the FA-DOX-magnetic nanochains in physiological saline solution, it was indicated that the nanochains, as exogenous substances, had an impact on the blood-initiating coagulation process, which was positively dependent upon the concentration. However, the addition of heparin can greatly prolong the hemoglutination time to over 60 min. Moreover, blood routine testing indicated that the nanochains had insignificant damage to normal leucocytes, erythrocytes, and platelets. Thus, the magnetic nanochains are suitable for the transvenous administration.

The magnetic separation of the magnetic nanochains was then verified. The experimental scheme was shown in Figure 5a and detailed intubation on the rat is shown in Appendix A. Normal rats with 250–350 g body weight and 8–10 mL circulating blood volume can tolerate the extracorporeal circulation. Here, the appropriate flow velocity of the bloodstream is a critical issue. It must ensure that the nanochains should be well mixed with cells but easy to separate, and the blood circulation should be safe. The final flow velocity of the bloodstream and the administration rate of nanochains were set at 0.3–0.5 mL/min and 0.1 mL/min, respectively. With the iterative blood circulation, the nanochains were increasingly separated from the blood and were reserved inside a micro-reservoir, which can be collected for recycling after the treatment. The separation efficiency *η* was calculated by $\eta =\frac{m}{m_{0}}\times 100\%$, where $m_{0}$ is the mass of Fe before the separation process, and *m* is the mass of Fe after separation process. The content of Fe was also determined by phenanthroline spectrophotometry. By calculation of Fe element alteration, the current separation efficiency was 60.41% (Figure 5b).

It was estimated that the majority of nanochains can be expelled from the body. With this clearance operation, the main organs have not been impacted significantly as shown from HE stained pathological sections (Figure 5c). The Fe element in these organs with different treatment was created with Prussian Blue staining (Figure 5d). It was found that under bright-field microscopy, there was no blue blot for the organs in the ECC group. For the MNC group, obvious blue blots appeared in the liver, the spleen, and the lung, and a few blots in the heart. This indicated that the nanochains can be intercepted by the organs during circulation. However, after the magnetic separation, the accumulation of Fe in the liver and the spleen significantly decreased, and there was even no accumulation in the lung and the heart. Moreover, in the presence of the rotational magnetic field, the magnetic separation could also effectively reduce the accumulation of nanomaterials in the organs. For the RMF group, there remained only a small number of blue blots in the liver. Therefore, the strategy of extracorporeal circulation and magnetic separation has been proven to be safer for clinical application in the future.

In cells, mechanical forces play a key role in affecting cell behaviors, including adhesion, differentiation, migration, and death. The magnetic field can control anisotropic magnetic nano-chains or nano-rods to produce magnetic force; this biomagnetic torsion pendulum has attracted extensive attention. Wilhelm et al. found that after a magnetic rotating stimulation, anisotropic magnetic nanorods were able to rotate freely for multiple rotations [40]. Cheng et al., reported that under the frequency rotating magnetic field, magnetic nanoparticles (MNPs) were able to form elongated aggregates with the size required to produce the elevated mechanical forces, and the internalized MNPs generated hundreds of pN to dramatically damage the plasma and lysosomal membranes by the physical disruption, leading to programmed cell death and necrosis [29]. Recently, they also revealed that by means of the RMF, the nanocubes assemble in alignment with the external field and produce a localized mechanical force to impair the cancer cells, which, in both in vitro and in vivo studies, damage the cancer cells and reduce the brain tumor growth rate after the application of the RMF. These studies demonstrate that the mechanical force generated by MNCs can exacerbate tumor cell damage upon the rotating magnetic field. We take advantage of the mechanical force controlled by MNCs in response to magnetic fields and their drug-loading properties, and this strategy contributed to desirable effects in the injury of leukemia cells, which provided a useful tool for dialysis clearance of leukemia cells in peripheral blood.

## 4. Conclusions

In summary, we proposed a novel approach to eradicate leukemia cells by the synergistic action of cytomembrane injury resulting from mechanical force and magnetic drug-loaded nanochains targeting the cancerous cells. Particularly noteworthy is that the elimination of leukemia cells was realized during the extracorporeal circulation and that the nanochains were magnetically separated after the treatment. This therapeutic approach greatly reduced the accumulation of nanomaterials in organs, further improving the safe application of nanochains in vivo. The ex vivo circulation-based system was subsequently proved safe on rats, and the magnetic nanochains showed little negative effect on the normal blood cells. The preliminary results demonstrate the potential to combat leukemia, thereby opening the door for a new therapeutic approach.

## Figures and Tables

**Figure 1 cells-11-02007-f001:**
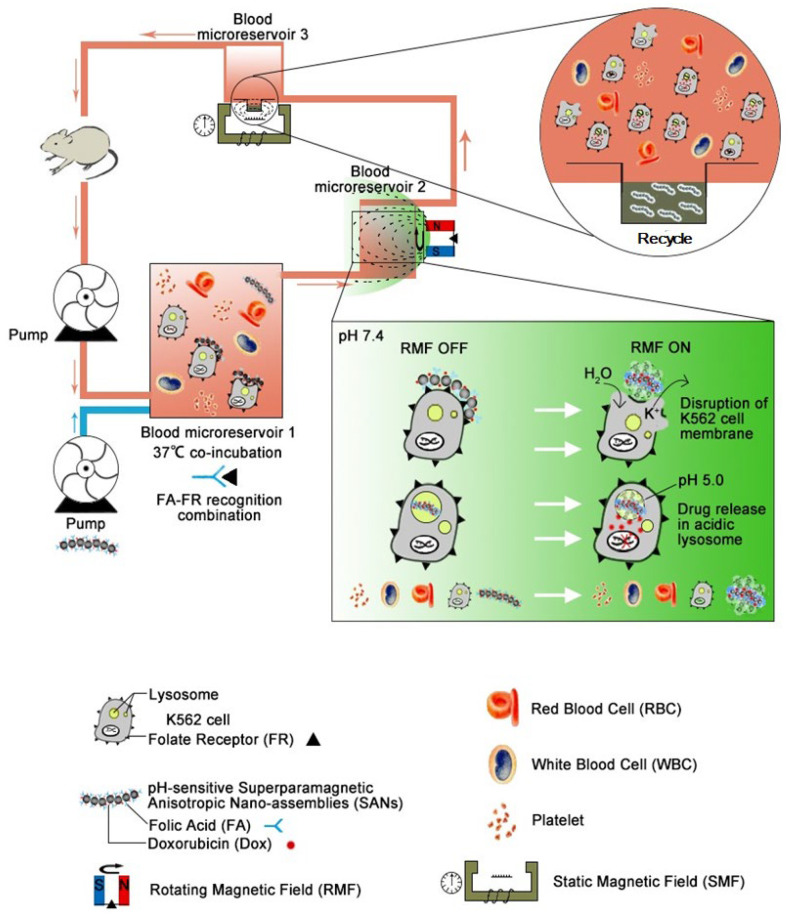
Schematic diagram of the nanochains used to eliminate the leukemic cells during the extracorporeal blood circulation.

**Figure 2 cells-11-02007-f002:**
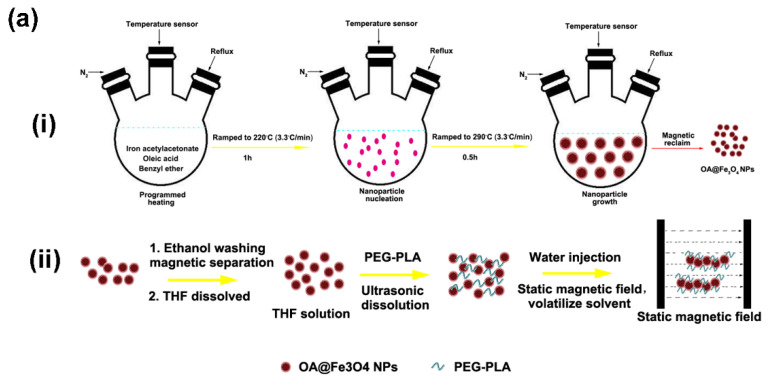
Construction of magnetic chain-like nanochains. (**a**) Schematic diagram of fabrication process by the solvent exchange method and the magnetostatic field-directed assembly. (**b**) TEM image of the original oleic acid@Fe_3_O_4_ nanoparticles. (**c**) TEM image of the fabricated magnetic chain-like nanochains. (**d**) The hydrodynamic size of magnetic nanochains in pure water, PBS and 10% serum medium at 0, 8 and 24 h, respectively. (**e**) Drug release profile of the nanochains in acidic and neural media, respectively. (**f**) Concentration dependent cytotoxicity to K562 cells of nanochains and FA-modified nanochains for 24 h. (**g**) Endocytosis of K562 cells to nanochains and FA-modified nanochains at four, 12, and 24 h, respectively. Data are presented as mean ± SD.

**Figure 3 cells-11-02007-f003:**
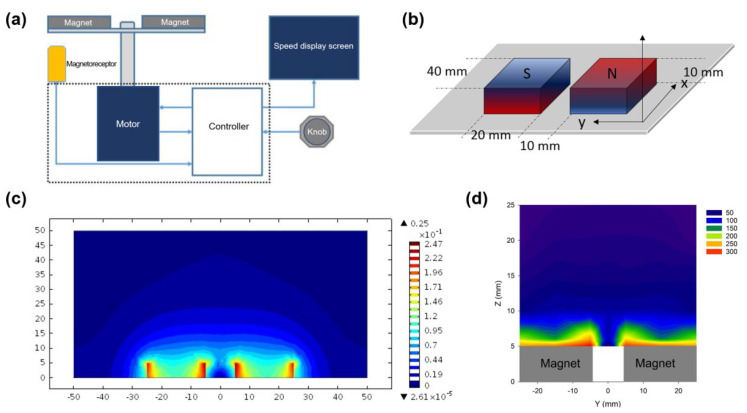
Rotational magnetic field for actuation of the nanochains. (**a**) Block diagram of the rotational magnetic field. (**b**) Model of the pair of magnets for simulation. (**c**) Contour map in the yz plane of the simulated magnetic field (Unit: Tesla). (**d**) Contour map in the yz plane of the magnetic field obtained by experimental measurement (35 positions totally at −25, −15, −5, 0, 5, 15, 25 mm on the *y* axis and 5, 6, 10, 15, 20 mm on the *z* axis, respectively).

**Figure 4 cells-11-02007-f004:**
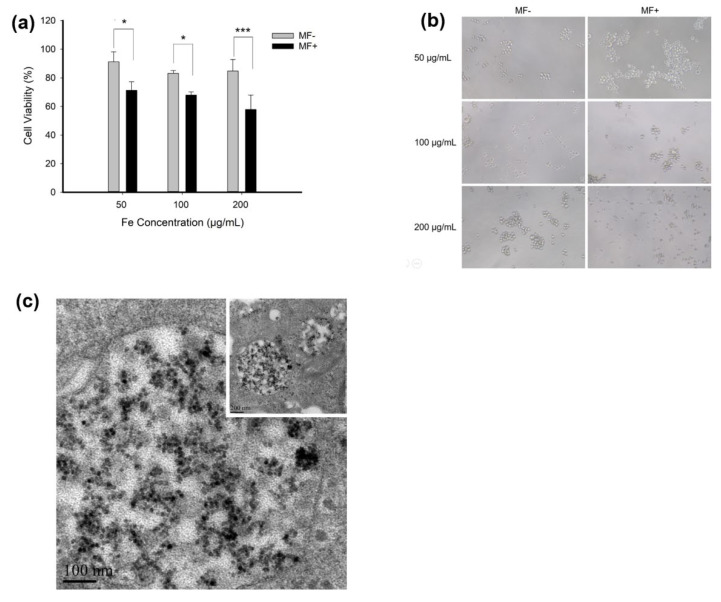
Killing of leukemic cells in vitro by the magnetic nanochains actuated by rotational magnetic field. (**a**) Rotational magnetic field significantly reduced the survival rate of K562 cells. (**b**) Morphological alteration of K562 cells treated by rotational magnetic field. (MF− means there is no magnetic field, while MF+ means there is a magnetic field. Scale bar: 50 μm). (**c**) TEM image of intracellular distribution of magnetic nanochains. Inset is the local magnification. (**d**,**e**) TEM images of lysosomes containing magnetic nanochains in the absence (MF−) and the presence (MF+) of the magnetic field, respectively. Under the treatment of the magnetic field, the nanocarriers can be seen in the cytoplasm, meaning the lysosomes were partially destroyed. (**f**) Apoptosis of K562 cells shown by Calcein-AM/PI staining. The number on the right was the ratio of dead cells. (FA-MNCs represent magnetic nanochains modified with folic acid. FA-MNCs-Dox represent doxorubicin-loaded folic acid modified magnetic nanochains. Scale bar: 100 μm). (**g**) Cell viability of Dox, FA-MNCs-Dox MF− and FA-MNCs-Dox MF+ for K562 cells were calculated based on OD values of CCK-8, and Dox content in the Dox group was the same as that in FA-MNCs-DOX (MF- means there is no magnetic field, while MF+ means there is a magnetic field). Data are presented as mean ± SD, *, *p* < 0.05, ***, *p* < 0.001.

**Figure 5 cells-11-02007-f005:**
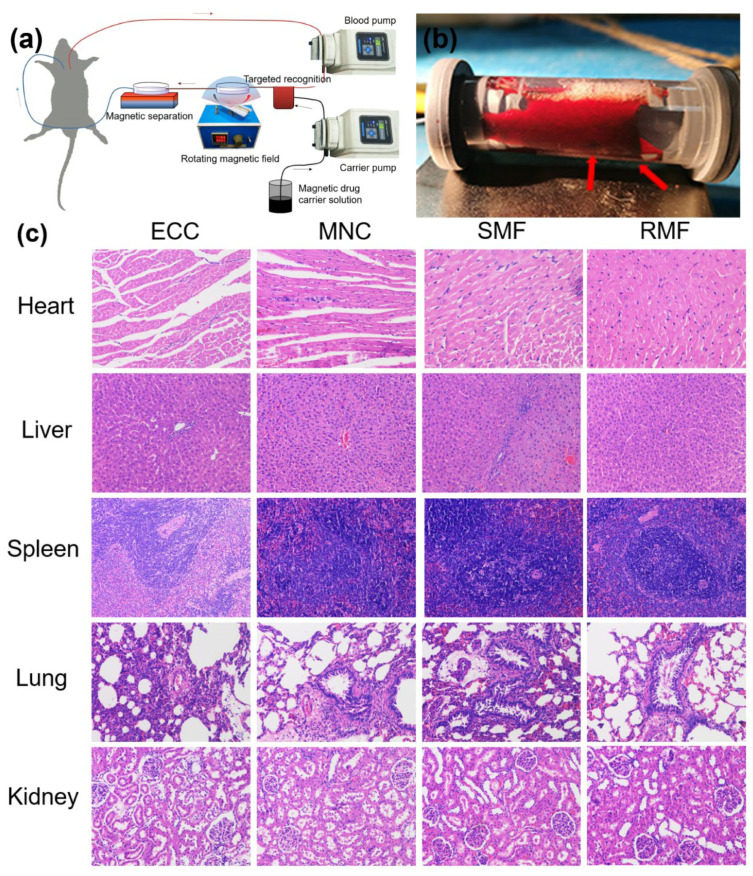
In vivo distribution of the nanochains on rats with the strategy of extracorporeal circulation plus magnetic separation. (**a**) Experimental scheme of the extracorporeal circulation plus the magnetic separation. (**b**) Exterior appearance of the magnetic separation chamber. Black sediments indicated by the red arrow were the separated magnetic nanochains. (**c**) HE staining images (×200) of main organs (rat) in different groups. (**d**) Prussian blue and nuclear fast red staining images (×200) of main organs (rat) in different groups. (ECC: Extracorporeal circulation, 2–3 mL of heparin sodium saline solution was pumped in; MNC: Magnetic nanoparticle chain, 2–3 mL of FA-MNCs-Dox heparin sodium saline solution was pumped in; SMF: Statistic magnetic field, 2–3 mL of FA-MNCs-Dox heparin sodium physiological saline solution was pumped in, and a static magnetic field was applied for carrier recovery; RMF: Rotating magnetic field, 2–3 mL of FA-MNCs-Dox heparin sodium physiological saline solution was pumped in. A rotating magnetic field was applied and a static magnetic field was used for carrier recovery).

## Data Availability

Not applicable.

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
