# Peer review of "Some Preliminary Results to Eradicate Leukemic Cells in Extracorporeal Circulation by Actuating Doxorubicin-Loaded Nanochains of Fe3O4 Nanoparticles"

_cells, 2022, doi:10.3390/cells11132007_

Round 1

Reviewer 1 Report

The authors resubmitted the remodeled version of the manuscript.

It is nice the authors provided some of the missing controls, it's a pity these controls are not present on the same figure (Figure 4 in the text). 

The authors did not provide the answer to question 2. As they did not have a mixed cell population all the text is just speculation.

The authors include the HD of particles upon coincubation in PBS with 10% FBS. To be more relevant, the HD should be measured in their blood mixture. 

Overall, in the beginning of the manuscript, the authors talk about the concept of their approach, then they describe the synthesis of the particles, subsequently they show the killing of cells in culture, and at last, they proved they can take out the nanochains by magnetic separation, once the nanochains are admixed to blood.

No results support that this approach could be made in vivo, nor that this approach might actually be efficient for leukemia. I re underline they made only one test in vitro on a given cell line. The manuscript presents some partial results that are connected with a story that is based on hypothesis and not on relevant scientific data.

Reviewer 2 Report

The manuscript improved but it is still hard to follow, I am coming back to the methods to follow the results, and so on. The authors resolved most of the issues, but there are still some modifications that require their attention:

  • did the authors compare the results of eradicating the cancer cells with the results from the literature using iron oxide-dox therapy? I keep trying to understand if this method is feasible, taking out the patient blood is a invasive method, the severity of the experiment should be taken into consideration;
  • did the ethic committee considered it severe?
  • so 1.2 % is a low incorporation rate and as mentioned before the S7 figure is the calibration curve of Dox used to determine the release profile;
  • what the authors mean by the reuse term from Figure 1?
  • revise also the Supporting file, is nanorobots there.

Reviewer 3 Report

The authors have appropriately revised the manuscript according to the comments, so I recommend the publication of this manuscript.

Author Response

Thank you very much for your comments.

Reviewer 4 Report

This manuscript introduces an innovative idea of adding nanochains were into the peripheral blood during extracorporeal circulation while subjecting it to a rotational magnetic field for actuation. Moreover, the leukocytes of lesion were conjugated by the nanochains via folic acid (FA) targeting. Finally, rotational magnetic field actuated the nanochains to release the drugs and effectively damage the cytomembrane of leukocytes.

The work is novel and well designed.

I recommend its acceptance for publication after carrying the following:

1- In the introduction: The authors should highlight the merits of the stimuli-responsive nanocarriers and state their different types. The authors can make use of: 10.3390/ma11071123 and 10.1016/j.jddst.2018.07.002.

2- Please elaborate the effect of the rotational magnetic field on the normal cells.

3- Please determine the nature of error bars in Figures 2 and 4

4- In the supporting information: All the figures mention the term "nanorobots" though this term was not mentioned at all in the main manuscript. Please justify.

Round 2

Reviewer 1 Report

The authors tried to answer the questions, but are unable to perform additional experiments due to Covid situation. In the same way as them, I highly value science and encourage high quality publications.

Please put figures 4 and S10 next to each other, not merged but in the same place, so the readers can get the information at one place. You can state in the caption that the experiments were not done at the same time.

In respect to the response to question 2, the authors mention low surface opsonization of PEG modification. As they have pegylated nanoparticles, and expose cells in extracorporeal flow in the dialysis like setting, how likely is that the leukemia cells would internalize the nanoparticles? For lysosomes to be disintegrated by nanoparticles rotation, the nanoparticles should first be internalized by the cell, but the authors state the chains are pegylated and we all know PEG provides stealth properties and lowers internalization. Please describe in detail how do you expect chains to work in this setting, would the cells internalize the chains and get destroyed. If we have adherent cells, cells adhere to the support and chains can rotate, generating torque and deforming the lysosomes. If cells are in suspension, such as leucocytes, wouldn’t the cells just turn on their selves, and thus the lysosomes would not disintegrate as there would be no cell resistance to the movement? Please comment.

On page 2, text highlighted in yellow, the authors mention stimuli responsive polypyrrole films and nanoparticles. What is the connection? If their idea is to provide a general overview on stimuli-responsive NPs, they should not just refer to polypirrole, rather provide a review with different stimuli-responisve NPs.

Author Response

Question 1:The authors tried to answer the questions, but are unable to perform additional experiments due to Covid situation. In the same way as them, I highly value science and encourage high quality publications.

Please put figures 4 and S10 next to each other, not merged but in the same place, so the readers can get the information at one place. You can state in the caption that the experiments were not done at the same time.

Response: thank you for noting this. We have put figures 4 and S10 next to each other.

Question 2: In respect to the response to question 2, the authors mention low surface opsonization of PEG modification. As they have pegylated nanoparticles, and expose cells in extracorporeal flow in the dialysis like setting, how likely is that the leukemia cells would internalize the nanoparticles? For lysosomes to be disintegrated by nanoparticles rotation, the nanoparticles should first be internalized by the cell, but the authors state the chains are pegylated and we all know PEG provides stealth properties and lowers internalization. Please describe in detail how do you expect chains to work in this setting, would the cells internalize the chains and get destroyed. If we have adherent cells, cells adhere to the support and chains can rotate, generating torque and deforming the lysosomes. If cells are in suspension, such as leucocytes, wouldn’t the cells just turn on their selves, and thus the lysosomes would not disintegrate as there would be no cell resistance to the movement? Please comment.                      

 Response: Thank you! Firstly, the number and type of exposed cells in extracorporeal flow are the same as those in blood circulation system. Secondly, Secondly, abundant literature have confirmed the targeting treatment of tumor by FA modification because of high expression of FA receptor in cancer cells. Just as the reviewer 1 rightly mentioned, PEG modification in common drug delivery system can diminish the immune clearance by RES(reticulo endothelial system). In our experiments, the role of PEGylation is to avoid nonspecific uptake of our carriers by immunocytes in blood, such as macrophages. Meanwhile, due to the FA modification, the leukemia cells can significantly internalize the nanochain carriers via the receptor-mediated internalization. Thus, the targeting clearance of leukemia cells was realized.The continuous circulation in situ could help enhance the collision probability between chains and all kinds of cells of blood. That is why we design a largest compartment for incubation room (Blood microreservoir 1). Of course, the internalization efficiency of suspension cells is not as so good as that of adherent cells.

Question 3: On page 2, text highlighted in yellow, the authors mention stimuli responsive polypyrrole films and nanoparticles. What is the connection? If their idea is to provide a general overview on stimuli-responsive NPs, they should not just refer to polypirrole, rather provide a review with different stimuli-responisve NPs.

Response: The added paragraph was actually written in response to the reviewer 4’s question about adding some literature. The reviewer 1 was right and we have provided a short review of different stimuli-responsive NPs. [Line 51-56]

Reviewer 2 Report

The comments were resolved, I would take out A new method and replace it with preliminary results according to the obtained results. 

Author Response

The comments were resolved, I would take out A new method and replace it with preliminary results according to the obtained results. 

Response: Thank you! We have changed the title from “a new method...” to “Some preliminary results...”

Reviewer 4 Report

The manuscript can be accepted in the current form.

Author Response

Thank you very much for your comments